

# A User-friendly Earth System Model of Low Complexity: The ESCIMO system dynamics model of global warming towards 2100

Jorgen Randers, Ulrich Golüke, Fred Wenstøp, Søren Wenstøp

BI Norwegian Business School, Nydalsveien 37, 0484 Oslo, Norway

*Correspondence to*: Jorgen Randers (jorgen.randers@bi.no)

**Abstract.** We have made a simple system dynamics model, ESCIMO, which runs on a desktop computer in seconds and is able to reproduce the main output from more complex climate models. ESCIMO represents the main causal mechanisms at work in the Earth system and is able to reproduce the broad outline of climate history from 1850 to 2015.

We have made many simulations with ESCIMO to 2100 and beyond. In this paper we present the effects of introducing in 2015 six possible global policy interventions that cost around 1,000 billion US$ per year – around 1 % of world GDP. We tentatively conclude a) that these policy interventions can at most reduce the global mean surface temperature – GMST – by up to 0.5°C in 2050 and up to 1.0°C in 2100 relative to no intervention. The exception is injection of aerosols in the stratosphere, which can reduce the GMST by more than 1.0°C in a decade, but creates other serious problems. We also conclude b) that

relatively cheap human intervention can keep global warming in this century below +2°C relative to preindustrial times. Finally, we conclude c) that run-away warming is unlikely to occur in this century, but is likely to occur in the longer run. The ensuing warming is slow, however. In ESCIMO, it takes several hundred years to lift the GMST to +3°C over preindustrial times through gradual self-reinforcing melting of the permafrost.

We call for research to test whether more complex climate models support our tentative conclusions from ESCIMO.

**Keywords**: Climate change, Feedback loops, Carbon Capture, Deforestation, Aerosols, Volcanoes, Greenland ice, Albedo

**Summary**. We describe ESCIMO – a system dynamics simulation model – which is designed to make it simple and

inexpensive for policy makers to estimate the effects of various possible human interventions to influence the global mean surface temperature – GMST – in this century. ESCIMO is simple enough to run on a laptop in seconds, and to make it possible to understand what goes on in the model system. ESCIMO is one integrated model consisting of sectors that track i) global carbon flows, ii) global energy flows, and iii) global albedo change. We show that ESCIMO, although simple, is capable of recreating the broad outline of the global climate history from 1850 to 2015. We show that ESCIMO is also able to recreate

the main scenarios generated by more complex climate models, both for GMST and other variables such as ice cover, ocean acidification, heat flow to the deep ocean, and carbon uptake in biomass. We present the tentative results of a number of experiments with ESCIMO where we simulate the consequences of several possible human interventions and natural disasters.





One conclusion is that human interventions that cost less than 1 % of world GDP are at most able to lower the temperature rise in 2050 by up to 0.5°C and in 2100 by up to 1.0°C, and not much more even if implemented jointly. The exception is injection of aerosols in the stratosphere, which can reduce the GMST by more than 1.0°C in a decade, but creates other serious problems. A second conclusion is that relatively cheap human interventions can keep global warming in this century below +2°C relative

to preindustrial times. Finally, we conclude c) that run-away warming is unlikely to occur in this century, but is likely to occur in the longer run. The ensuing warming is slow, however. In ESCIMO it takes several hundred years to lift the GMST to +3°C over preindustrial times, through gradual self-reinforcing melting of the permafrost. We call for research to check whether more complex climate models support these tentative results from ESCIMO. ESCIMO is an acronym for Earth System Climate Interpretable Model.

**1 Research objective**

**1.1 A simple user-friendly model**

We have built a system dynamic simulation model that makes it simple and inexpensive for the user to calculate the effect of various possible human policy interventions intended to influence the global temperature in this century. The model is so simple that it can be run in seconds on an ordinary personal computer, and makes it possible to understand what goes on in the

model system. The model is named ESCIMO for "Earth System Climate Interpretable Model", and is a rare addition to the tiny group of earth system climate models of low complexity that exist today.

Our focus is global, and primarily on the "global mean surface temperature" (GMST) and other top-line macro descriptors of the climate system. Our focus is on the middle term – to the year 2100.

We have written this paper for two reasons. The first is to make the educated public, decision makers, and scientists aware that

they have the option to use simple models like ESCIMO to make climate-policy experiments on their own personal computer. The second reason is to encourage the teams that run complex climate models to test whether their models support our tentative conclusions from experimentation with ESCIMO.

**1.2 Positioning in literature**

A number of climate models exists that seeks to mimic selected characteristics of the global climate system. IPCC (2013,

pp.747, 748) gives a useful overview, listing 54 relatively complex models and 15 somewhat simpler models, so-called Earth Models of Intermediate Complexity (EMICs). ESCIMO is an addition to the last group of models. For more detail, see (IPCC, 2013, pp. 854-866).

Climate models represent fundamental laws of nature, and seek to reproduce historical observations. They are used to calculate future trends in response to possible human interventions. Over time, climate models incorporate an ever-increasing number

of components to describe the real world mechanisms that influence the global temperature more precisely. ESCIMO follows in this tradition, but at the same time, we seek simplicity – to obtain a transparent, understandable, causal and dynamic model.



Much work is being done to assess the quality of climate models, primarily by comparing model outputs with historical data (IPCC, 2013, Ch. 9, pp. 741), and also by comparing future trends generated by one model with the trends from other models using the same input assumptions. While one single model will never be capable of reflecting the actual climate system and its inherent dynamics perfectly, the models in use tend to get increasingly accurate over time (IPCC, 2013, Ch. 9 and 12). We

subject ESCIMO to the same quality assessment as existing models, but we have the additional ambition of presenting a simple and aggregate perspective on the global climate system. ESCIMO differs from many models in being integrated and dynamic, and includes Earth system processes that may be important, even if their strength is still very uncertain.

IPCC distinguishes between three groups of models: a) Simple Energy Balance Models which are not dynamic, b) Atmosphere-Ocean General Circulation Models (AOGCMs) which are dynamic, but do not include biogeochemical feedbacks, and c) an

increasing number of state-of-the-art Earth System Models (ESMs). Of these, the Earth System Models of Intermediate complexity (EMICs) tend to be more comprehensive, but with a lower resolution than the most complex models. ESCIMO belongs to the EMICs, but with a still lower complexity and may be classified as an Earth System Model of Low Complexity (EMLC). Most models are complex because they are spatially compartmentalized and consist of linked modules that must be iterated towards a solution. None, with the exception of (2015a) are sufficiently simple to allow users to conduct their own

experiments within a few minutes (Sterman et al., 2012). This is our ambition with ESCIMO.

A crucial question is whether such a simple model is capable of producing useful policy advice. Our ambition is to show that a model can be simple at the same time as it is comprehensive and useful. To be comprehensive, a model of the climate system must include the central components of the climate system as well as its main dynamics. To be useful, it needs to be able to provide quick and understandable answers to questions of interest to users.

**1.3 Intended use**

Our prime goal is to demonstrate that it is possible for interested persons to use their own computers to calculate the climate effect of various policy proposals. They do not have to rely on simulations by specialist teams running the more complex models that currently dominate the scene. We want them to know they have the option to use models like ESCIMO, which is simple and cheap. To this end, we have made ESCIMO freely available on the web, as described at the end of this article.

In a sense, ESCIMO represents a "disruptive technology": a very much cheaper way for politicians and others to estimate the likely effect of proposed climate policies. Needless to say, complex models are still needed to calculate the finer detail, and to ensure that the simple models are not misleading.

**1.4 Improved understanding**

Our second goal is to make a model that is simple enough to be understandable – in the sense that it is possible to explain why

something happens in a model run.

The scope of ESCIMO is very broad since we want to make it possible to compare the effects of a wide range of human interventions. Examples are reduction of $CO_2$ emissions from energy production, changing forestry practices, increasing



Earth's surface albedo, as well as various geoengineering proposals. We also want ESCIMO to help users understand the effect on temperature of natural disasters like volcanic eruptions, accelerated glacier melting, methane burps from the permafrost, methane clathrate release and so on.

## 2 Model description

ESCIMO is our summary of the available literature on the climate system, its structure, parameter values, and behavior. The model is a set of around 50 non-linear differential equations that constitutes a top-down representation of the main causal processes that influence the evolution of GMST. ESCIMO is largely consistent with the "IPCC view of the world". It consists of sectors that track i) global carbon flows, ii) global energy flows, and iii) global albedo change. The system's evolution over time is determined by a number of endogenous interlinked causal processes, all gathered in one integrated model.

Below, we present ESCIMO in four different ways: as a high-level sector diagram, as a set of feedback loops, as a system of stocks and flows, and as a list of the biogeochemical processes included. A complete list of the equations and the parameter values is available on the web as described in the final section of the article.

### 2.1 High-level sector diagram

ESCIMO consists of three main sectors (see Fig. 1): Carbon, Energy, and Albedo, with a number of important interlinkages.
The Carbon sector tracks the flow and distribution of carbon in the earth system, both in the form of $CO_2$, $CH_4$ and $C$ – in the atmosphere, biomass, ocean water, and sediments. The Energy sector tracks the flow and distribution of energy, both in the form of light (short wave radiation) and heat (long wave radiation), in the atmosphere, the surface, and the deep ocean. The Albedo sector tracks the changes in albedo (reflectivity) that arise when the surface shifts among three main categories: very bright (covered by snow and ice), bright (tundra, deserts, savannah), and dark (forests, ocean, barren land). The Albedo sector
also tracks the extent of net-warming high clouds and net-cooling low clouds. ESCIMO would have been still better if it also included a complete Water sector, tracking the flow of $H_2O$ in the Earth system. In the current version of ESCIMO, we only track the important parts of the water cycle in atmosphere, clouds, and on-land ice.





**Figure 1. The three sectors in ESCIMO**

## 2.2 Major feedback loops

ESCIMO contains eight major feedback loops that interconnect the three main sectors and drive the overall dynamic time
development of the model system (see Fig. 2). The eight loops are described below. There are many other loops in ESCIMO,
but these eight explain most of the dynamic behavior of the model system up to the year 2100. In Fig. 2, we use conventional
system dynamics notation to indicate physical stocks with boxes and physical flows with valves. Causal links are shown as
simple arrows, and the polarity of the links with minus signs for opposite, and plus signs for the same polarity. To avoid
cluttering, the pluses are omitted in Fig. 2, only the minuses are shown. Polarities of the loops are also shown, with minus for





balancing and plus for self-reinforcing loops, and placed in front of the loop number. Finally, a long delay involved in the operation of a loop is indicted with two short parallel lines (‖) crossing the relevant causal link.

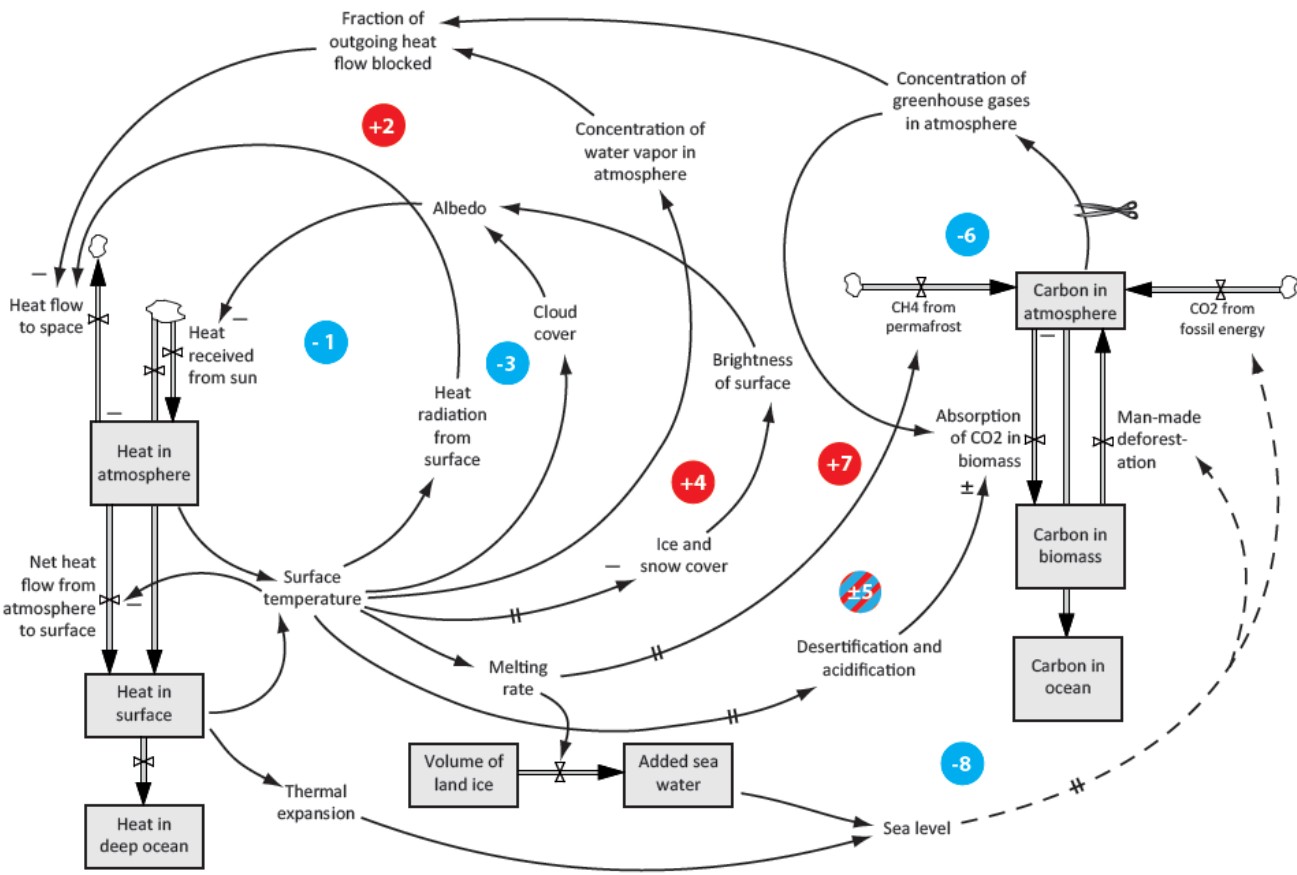

**Figure 2. The eight major causal loops in ESCIMO. A plus sign in front of a loop number identifies a "self-reinforcing loop", while a minus sign identifies a "balancing loop".**

Loop 1: A higher surface temperature GMST leads to higher heat radiation from the Earth's surface, and higher heat flow to space, which in turn reduces the amount of heat in the atmosphere and lowers GMST. This is a balancing (negative) feedback loop, and strong, as the outgoing heat flow increases with the fourth power of the temperature, according to Stefan-Boltzmann's law.

Loop 2: Warming leads to a higher concentration of water vapor in the atmosphere, which, in turn, blocks a larger fraction of the heat that escapes into space. The result is more heat in the atmosphere and higher GMST. This is a self-reinforcing (positive) feedback loop, but with diminishing gain in the operating range of the model and the climate system. The saturation level for water vapor in the atmosphere is a function of temperature only, and water vapor therefore acts to amplify – and maintain –





the initial temperature effect of adding man-made greenhouse gas to the atmosphere. The loop is fast and its effect of long duration.

Loop 3: Warming increases the area covered by low clouds, which in turn leads to higher reflection of the incoming sunlight, with a cooling effect. This is a balancing feedback loop, but with low gain, as the cloud cover only increases slowly with

increasing GMST. High clouds have a warming effect, but it is ten times weaker than the effect of low clouds.

Loop 4: Warming leads to melting of ice and snow, exposing underlying darker ocean and rock, which, in turn reduces the albedo and leads to higher absorption of incoming sunlight, and increased warming. This is a self-reinforcing feedback loop, but other long-term effects of warming on the surface cover (e.g. replacing dark forests with brighter deserts) reduce the strength of this feedback loop. The strength declines to zero once all ice and snow has melted.

Loop 5: Warming increases the rate of biomass growth through photosynthesis on land and in the ocean, which in turn reduces the amount of $CO_2$ in the atmosphere. This reduces the concentration of greenhouse gases in the atmosphere, which in turn blocks a smaller fraction of heat that escapes into space. The result is reduced warming. This is a balancing feedback loop, which is strengthened by the fertilization effect of $CO_2$ on biomass growth (see Loop 6), and weakened by acidification of the oceans and by desertification of former forests and grasslands. The latter effects are sometimes strong enough to transform

Loop 5 into a self-reinforcing loop.

Loop 6: More carbon in the atmosphere leads to increased absorption of $CO_2$ in terrestrial biomass through fertilization, which, in turn, leads to less greenhouse gases in the atmosphere, and less absorption in terrestrial biomass. This is a balancing feedback loop, which works as long as $CO_2$ is an important determinant of biomass growth. In parallel, more carbon in the atmosphere leads to more $CO_2$ in the ocean surface layer – through chemical diffusion – and in turn faster removal of $CO_2$ from ocean

surface water through assimilation in aquatic biomass and sedimentation. This process is slowed – and ultimately reversed – when the concentration of $CO_2$ in the ocean surface layer increases and the water becomes more acidic.

Loop 7: Warming melts permafrost which releases $CH_4$ (methane) and $CO_2$ from thawed organic material. This increases the concentration of greenhouse gases in the atmosphere, which in turn blocks a bigger fraction of heat that escapes into space. The result is increased warming. This is a self-reinforcing feedback loop, strong at first, but the gain goes to zero once all

permafrost has melted.

Loop 8: Warming increases the sea level by the melting of on-land ice and snow. Warming also leads to thermal expansion of the ocean water, first in the top layers. This far, humanity has simply adapted to changing sea levels. In the future, there is the possibility that a rising sea level will eventually make humanity reduce emissions of greenhouse gases, which in turn would lead to reduced warming. This would close Loop 8 as illustrated with the dotted link, and make it a balancing feedback loop

that will gain strength when the increasing sea level becomes a serious problem for humanity. However, the dotted link is not included in the current version of ESCIMO, and hence Loop 8 is not operational in the simulations described later in this paper.



## 2.3 Stock and flow diagram

Each of the three sectors has an internal network of stocks and flows that ensures conservation of carbon, energy and surface area. Figure 3 shows the stock and flow diagram for carbon. The diagram illustrates the proportions of the stocks of carbon (in GtC) and of the main flows of carbon (in GtC/yr) among the stocks at a given point in time. 1 Gt equals $10^9$ tons. Most of the

5   carbon resides in the ocean. There are also significant stores in biomass (plants and soil), and in permafrost (frozen biomass). Little resides in the atmosphere, only one tenth of what is stored in reserves of fossil coal, oil and gas.

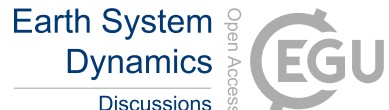

**Figure 3. Stocks (inGtC) and flows (in GtC/yr) in the Carbon sector of ESCIMO. World around year 2000.**



Figure 4 gives the same picture for the stocks (in ZJ) and flows (in ZJ/yr) of energy. 1 ZJ equals $10^{21}$ J, and 1 J equals 1 Ws. Note that an energy flow of 1 W onto each $m^2$ of the total surface of the earth equals some 16 ZJ/yr. Almost all the energy is stored in the ocean and very little in the land surface. Even today, with ongoing global warming, the incoming and outgoing

energy flows are almost in perfect balance. 340 W/$m^2$ is shining in from the sun, while 338 W/$m^2$ is radiating back into space







**Figure 4. Stocks (in ZJ) and flows (in W/m$^2$ ≈ 16 ZJ/yr for whole earth) in the Energy sector of ESCIMO. World around year 2000.**





Finally, ESCIMO splits the total surface area into the following surface types: ocean, ice-covered ocean, ice-covered land, tundra, barren land, forest, grasslands, deserts and built land. The fractions sum to 100%, and all areas change with global warming, but so gradually that there is limited accumulated change before 2100 – except when it comes to ice-covered ocean, ice-covered land, and the area of permafrost.

## 2.4 Processes included in ESCIMO

The flows of carbon, energy and land surface types between the stocks are governed by a number of biogeochemical processes, and some are influenced by human activity. The following processes are included in ESCIMO in a causal manner. For each process, we describe its development over time, which arise as a result of drivers in other parts of ESCIMO. In most cases –

unless when noted as "always > 0" – the process can go both ways. For example, today the sea ice is melting, but it would refreeze if conditions changed sufficiently. If a process change direction, we use negative values for the flow. The processes are grouped according to IPCC's categories (IPCC, 2013, pp. 864-866). The whole system is driven by man-made emissions of $CO_2$, $CH_4$, and other greenhouse gas molecules to the atmosphere.

ATMOSPHERE

1. Flow of incoming energy (light and heat) from the sun (constant, with tiny oscillations due to an exogenous sunspot cycle, always >0)

2. Reflection of incoming sunlight from the top of high and low clouds (proportional to the clouds' extent, always > 0)

3. Reflection of incoming sunlight from the surface of land and ocean (proportional to the extent and albedo of various surface types, always > 0)

4. Blocking of outgoing heat (long wave radiation) by $CO_2$, $CH_4$, water vapor, and other greenhouse gas molecules (proportional to functions of approximately logarithmic shape of the concentration of the various gases, always > 0)

5. Radiation of heat from Earth into space (proportional to the surface temperature in °K to the fourth power, always > 0)

6. Re-radiation of heat from inside high and low clouds (proportional to the clouds' extent, always > 0)

7. Addition of water vapor to the atmosphere (the equilibrium water vapor concentration is a steeply rising function of temperature in °C)

8. Addition of high (net-warming) clouds (cloud extent is proportional to temperature in °K)

9. Addition of low (net-cooling) clouds (cloud extent is proportional to temperature in °K)

OCEAN

10. Heat transfer from atmosphere to land and ocean surface (several processes (see 15, 16, 17) driven by the temperature difference in °C)

11. Heat transfer from surface ocean to deep ocean (proportional to the temperature increase since 1850 in °C)



12. Sea level rise  (sum of the addition of water from melted land ice and the thermal expansion caused by higher ocean temperatures)

13. Carbon transfer from surface ocean to deep ocean (determined by the long term average  speed of down- and upwelling)

SEA ICE

14. Melting of sea ice (proportional to the surface temperature in °C)

COUPLING

15. Heat transfer from atmosphere to surface (proportional to the extent of clouds)

16. Direct heat transfer from atmosphere to surface (proportional to the fourth power of the temperature in the atmosphere
10       in °K

17. Heat transfer from surface to atmosphere  (sum of convection and evaporation (proportional to temperature in °C) and radiation (proportional to the fourth power of surface temperature in °K))

18. $CH_4$ transfer from permafrost to atmosphere (proportional to the rate of melting of the permafrost, always > 0)

19. $CO_2$ transfer from air to top surface layer of the ocean (a chemical diffusion process proportional to the concentration
difference)

20. Aerosols from volcanic activity (exogenous, always >0)

21. $CO_2$ from biomass released by forest fires (proportional to temperature in °C)

LAND SURFACE

22. Melting of permafrost  (proportional to the surface temperature in °C, as long as there is permafrost left)

23. Melting of snow and ice cover on land (proportional to the surface temperature in °C, as long as there is snow and ice cover left)

24. Melting of ice cover on ocean (proportional to the surface temperature in °C, as long as there is ice cover left)

25. Increase in the area covered by (bright) desert and savannah  (proportional to surface temperature in °C)

26. Reduction in the area covered by (dark) forest  (proportional to surface temperature in °C)

BIOSPHERE

27. $CO_2$ transfer from air to biomass on land - beyond the annual growth cycle (increases with the concentration of $CO_2$ and declines with the surface temperature)

28. $CO_2$ transfer from ocean to biomass in the ocean (increases with the concentration of $CO_2$ and declines with the ocean temperature, always > 0)

ICE SHEETS

29. Melting of glaciers on land  (same as process 21)

SEDIMENTS AND WEATHERING

30. $CO_2$ transfer from deep ocean to ocean sediments (proportional to the amount of $CO_2$ in the ocean, always > 0)



The level of detail in the description of these processes in ESCIMO varies, but in all cases, the equations embody a causal biogeophysical explanation of the relevant process.

## 3 Model simulations

### 3.1 Reproducing history – The base run from 1850 to 2015

The first test of the quality of ESCIMO is to check the extent to which the model is able to reproduce history. Figures 5a and b show the result:  The simple ESCIMO model structure, when parameterized with plausible parameter values obtained from the literature or common sense, and driven by actual man-made emissions of greenhouse gases from 1850 to 2015 (Figure 5b lower right), is able to reproduce the broad lines of the global climate history. The future portion of these graphs is generated by ESCIMO with what we see as the most likely man-made emissions from 2015 to 2100 (Figure 5b lower right). Figures 5a and b present what we call the "base run" from 1850 to 2100. They show the time development of a selection of model variables[1] in response to historical time series for man-made emissions of $CO_2$, $CH_4$ and the other Kyoto-gases, and for the incidence of volcanic eruptions.

---

[1] When running ESCIMO with the Vensim software, the user can  freely choose which variables in ESCIMO to study, from all (including  the roughly one hundred important) variables



**Figure 5a. ESCIMO base run (blue) compared to history (red). World 1850 to 2100.**



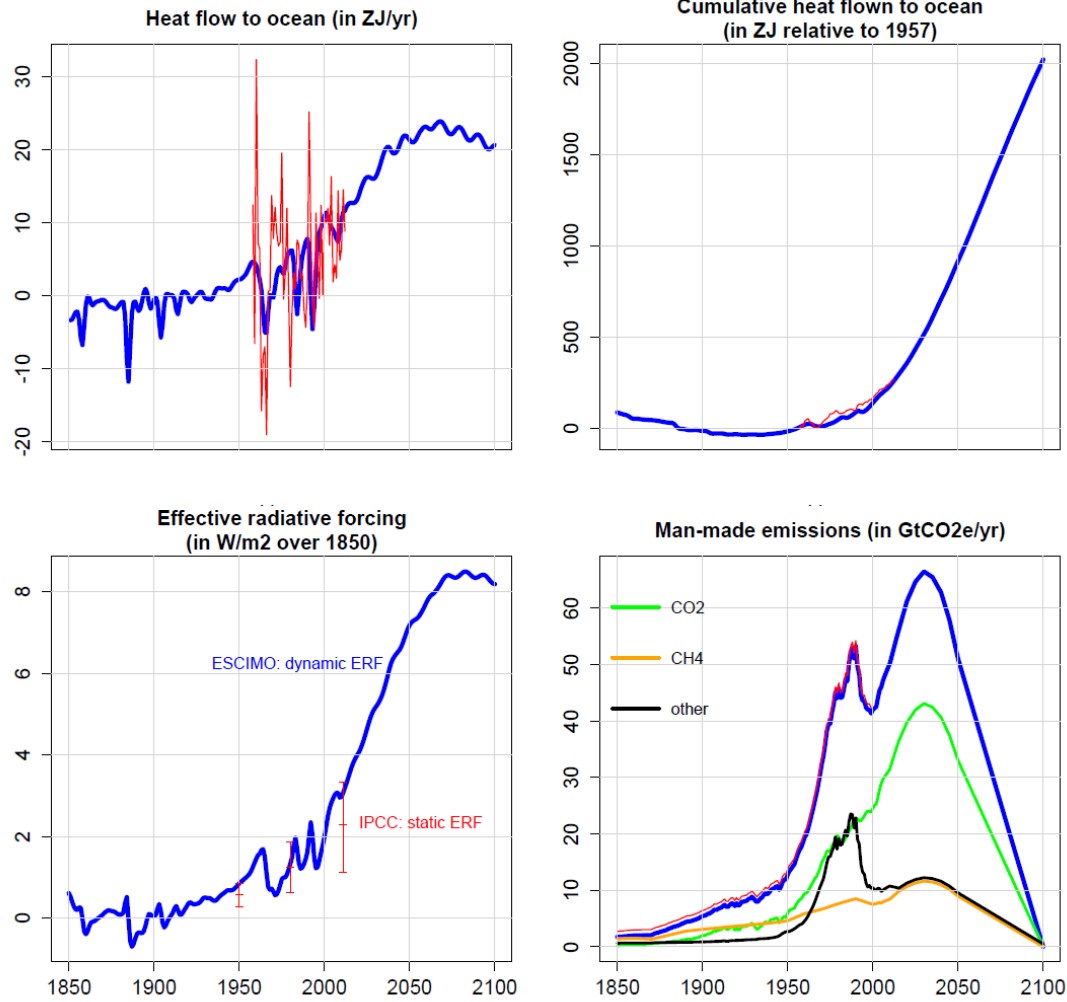

**Figure 5b ESCIMO base run (blue) compared to history (red). World 1850 to 2100.**

Figures 5a and b also show how the ESCIMO base run compare with the observed trends for a selection of variables for which
5   data are available:

   – Temperature rise (in GMST) – in °C since preindustrial times. Sources: (NOAA, 2015a, b)

   – Concentration of $CO_2$ – in ppm. Source: (Meinshausen et al., 2011)

   – Sea level rise – in meters relative to 1850. Source: (Church and White, 2011)

   – Arctic ice extent – annual average in million km². Source: (2015b)

10   – Acidity of top ocean layer – in pH. Sources: (Bates et al., 2012;Dore et al., 2009)

   – Concentration of $CH_4$ – in ppb. Source: (Meinshausen et al., 2011)

   – Heat flow from air to ocean – in ZJ per year. Source: (Levitus et al., 2012)





– Cumulative heat flow to ocean – in ZJ from 1850. Calculated from Levitus et al. (2012).

– Effective radiative forcing – in Wm$^{-2}$ relative to 1850. Source: (IPCC, 2013, p. 1535)

Figures 5a and b show that there is a reasonable match between the output from ESCIMO and the trends in the historical data. The only significant discrepancy is in "effective radiative forcing" ERF, but this is because ESCIMO generates the "dynamic"

5   real-time radiative forcing (the net incoming energy flux at top of the atmosphere), while IPCC reports on a "static" and more constrained concept.

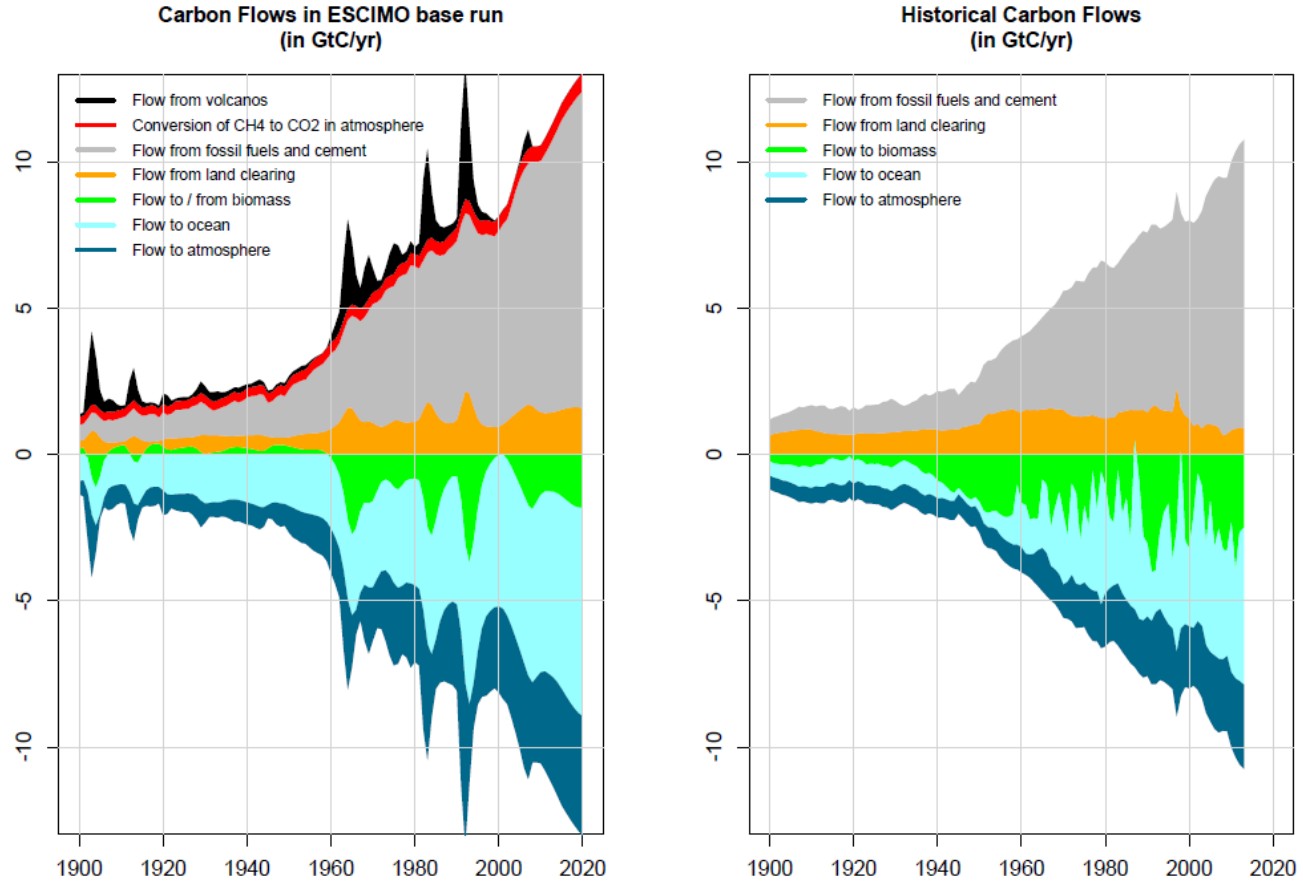

**Figure 6 Carbon flows (in GtC/yr) in the ESCIMO base run compared with observed carbon flows. World 1900 to 2015.**

Figure 6 shows that the base run roughly matches the historical development over time of global carbon flows (in GtC/yr)

10   from 1900 to 2015 (Le Quéré et al., 2015). Note that ESCIMO includes sources of carbon, for example from volcanic activity and conversion of CH$_4$ to CO$_2$, which are not included in the historical data (which are not "real data" but an assessment based on various pieces of information including some measurements).



### 3.2 Reproducing the RCP scenarios

A second test of the quality of ESCIMO is to explore how well the model is able to match the output from more complex models for the 2015 to 2100 period, when driven with the same inputs. The IPCC has defined four scenarios that represent four possible scenarios for man-made greenhouse emissions to 2100. They are called Reference Concentration Pathways –

5 "RCP"s – and are characterized by their (theoretically derived) radiative forcing in 2100 in W/m$^2$. In Fig. 7 we use the IPCC RCP 4.5 emissions scenario from 2015 to 2100  (Stocker et al., 2013;Ward et al., 2012) to drive ESCIMO. Figure 7 shows that ESCIMO generates a temperature scenario that falls well within those generated by the CIMP5 ensemble of complex models (2015c; Eschenbach, 2014), when driven with the same RCP 4.5 inputs.

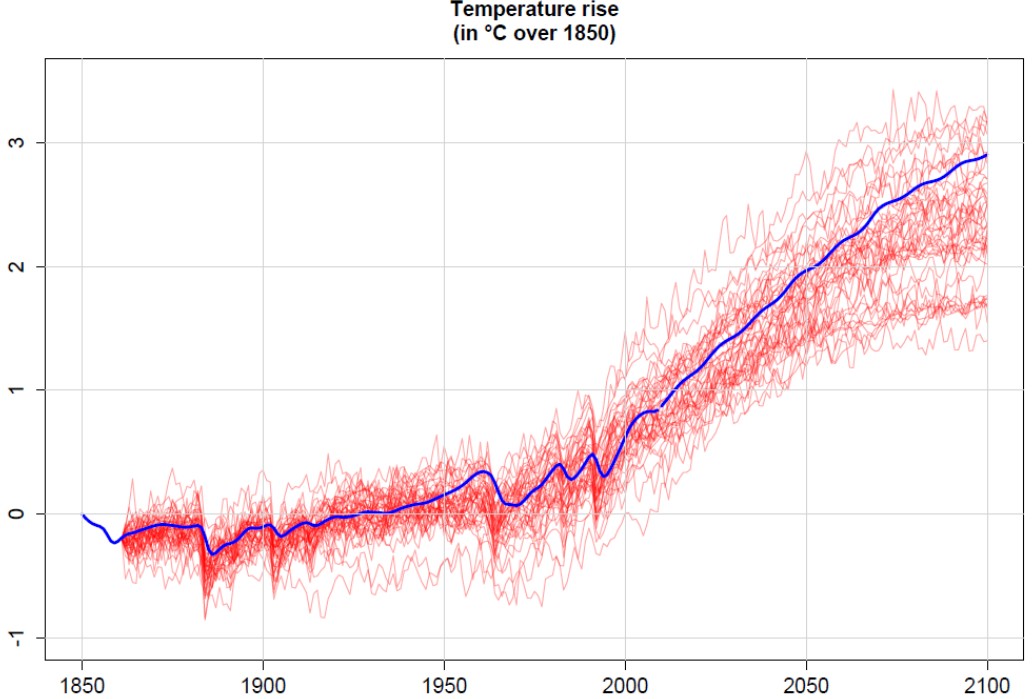

**Figure 7 Temperature rise from 1850 in  ESCIMO when driven by the RCP 4.5 emissions scenario (blue) compared to the output from 42 models (the CIMP5 ensemble) driven by the same emissions scenario (red). World 1850 to 2100.**

Figure 8 shows the results of additional tests of ESCIMO where it is driven by other inputs. To the left in Fig. 8 we show the result of driving ESCIMO with the four RCP emission scenarios. The ESCIMO base run fits nicely within the output generated

by these four scenarios, both when it comes to  temperature, sea level rise, and future emissions.





**Figure 8 The response in ESCIMO to four RCP emissions scenarios (left) and to four other experiments (right). All compared to the ESCIMO base run (blue). All changes introduced in 2015. World 1850 to 2100.**





In the top graph to the right in Fig. 8 we show the result of driving ESCIMO with two other emission scenarios: a) a sudden reduction to zero of man-made emissions from 2010, and b) constant man-made emissions from 2010. Again, ESCIMO generates outputs similar to the average of more complex models, when driven with the same inputs.

It is worth noting that ESCIMO generates a small increase in the temperature just after the introduction of the zero emissions policy in 2015, as do the more complex models. The reason is that the experimental setup cuts all emissions, including aerosols, to zero. This reduces the albedo (reflectivity) of the planet, which in turn increases the absorption of incoming solar energy.

### 3.3 Reproducing climate system descriptors

A fourth test of the quality of ESCIMO is to explore how well the model is able to match the more complex models regarding
important system descriptors or "performance metrics" (IPCC, 2013, Table 9.5, p. 818). Many of the commonly used descriptors are equilibrium concepts, which are emergent properties of a dynamic model like ESCIMO and only will arise in the real world after hundreds or thousands of years. Although ESCIMO can be run as long as one wants, such long runs are usually unrealistic since we have deliberately excluded all causal mechanisms that work on a longer timescale than a thousand years – like the effect of the lower gravitational pull on the ocean when the Greenland ice mass melts. The hypothetical
equilibria that form the conceptual basis for many of the performance metrics commonly used in climate modelling describe unrealistic situations that requires artificial cutting of feedback loops in the model system. In other words, these equilibrium metrics are measures of the mathematical characteristics of the models, and say little about the real world.

Still, we have estimated the values in ESCIMO of two commonly used metrics, namely the "equilibrium climate sensitivity" and the "transient climate response" – for which the IPCC has published high-level conclusions (IPCC, 2013, p. 1033). The
ESCIMO output is shown in the two lower graphs to the right in Fig. 8, and in both cases, it falls within the ensemble of results from more complex models.

In order to generate these two outputs, we have to cut the link from "Carbon in the atmosphere" to "Concentration of greenhouse gases in the atmosphere" at the place marked with a scissors in Fig. 2.

The "equilibrium climate sensitivity" ECS is defined as the temperature rise that will result in equilibrium after a doubling of
the atmospheric concentration of $CO_2$ (IPCC, 2013, p. 1451). To calculate the value of this metric in ESCIMO would have required us to continually inject enough $CO_2$ into the model atmosphere to compensate for the $CO_2$ that is continually being moved from the atmosphere into the ocean and biomass. Instead, we chose to hold the concentration artificially stable by cutting the causal link from the variable "Carbon in atmosphere" to the variable "Concentration of greenhouse gases in the atmosphere". Then we exogenously doubled the "Concentration of $CO_2$ in the atmosphere" at the end of 1850, kept it constant,
and let the temperature evolve in the system. The result is shown in Fig. 8 (middle right). The temperature rises by +3.1°C after 100 years, and by +3.7°C after 150 years. This is within the range of 1.5 to 4.5°C given as the equilibrium climate sensitivity of the more complex models (IPCC, 2013, p. 16).



The "transient climate response" TCR is defined as the temperature increase after 72 years of steady 1 % per year increase in the concentration of $CO_2$ in the atmosphere. Or, in other words, the temperature rise after a doubling of the concentration of $CO_2$ at a predetermined speed of 1 % per year. As shown in Fig. 8 (bottom right), ESCIMO generates the value +1.9°C, which is within the range of 1.0°C to 2.5°C produced by more complex models (IPCC 2013, p 16). To achieve this result, we once

more cut the causal link from "Carbon in the atmosphere" to "Concentration of greenhouse gases in the atmosphere", and then increased the $CO_2$ concentration by 1 % per year exogenously.

### 3.4 The most likely future – The base run from 2015 to 2100

We conclude that the output from ESCIMO not only is able to reproduce the broad outline of climate history from 1850 to 2015, but also to reproduce the broad outline of the output from more complex models when driven by the same exogenous

drivers. We now proceed to make experiments with the model – testing the effect of interesting changes in parameters or policies. For this, we need a starting point, a continuation of the base run from 2015 to 2100, with which we can compare our experimental runs. And for this in turn we need a scenario for the exogenous drivers over the 2015 to 2100 period – primarily a scenario for the man-made emissions of climate gases. We chose a scenario – a future emissions pathway – that we believe is more likely than any of the RCP scenarios. This is a future where man-made greenhouse emissions rise to a flat peak around

2030 and then gradually decline to zero in 2100. This future is described in detail by Randers (2012) and reflect the beliefs of the present authors.

Figure 5b (lower right hand graph) illustrates what we see as the most likely man-made emissions profile measured in $GtCO_2e$ per year. Figure 8 (top left) shows that our most likely scenario falls in the middle of the four RCP scenarios. Our base case emissions follow the RCP 8.5 scenario up to 2025 as global society keeps postponing a strong response to the climate challenge.

Then base case emissions gradually shifts towards the RCP 6.0 scenario as global society finally accelerates its effort to cut the use of fossil fuels. Finally, after 2080 base case emissions shift downwards to the RCP 2.5 scenario as global society finally gets rid of the last fossil energy and the climate-intensive agriculture and forestry. Many argue that ours is an optimistic scenario, others argue the opposite. As we see it, our base run emissions scenario reflects the decisions that are likely to be made and implemented in the world in the 21[st] century. Thus, our base run includes all "ordinary" future action, but not the

"extraordinary" actions that we test in next chapter.

The graphs in Fig. 5a and b show what we see as the most likely future – the result of driving ESCIMO with our base case emissions profile. GMST peaks at +2.4°C relative to preindustrial times around 2085, and declines slowly during the 2090s. In 2100, the sea level is 0.9 m above the level in 1850, or 0.6 m above the level in 2015. The acidity passes a minimum around 2050 and start moving back towards normal. Arctic sea ice is reduced by 40 % relative to 2000. By the year 2100, the ocean

has absorbed twice as much heat as there was in the atmosphere around 2000, and also one half of the amount of carbon that was in the atmosphere around 2000. Big volumes of heat and carbon are continuing to flow towards the deep ocean.

There are few surprises in the ESCIMO most likely future from 2015 to 2100. The system evolves as one would expect from the assumptions made. But there are two points worth making. First, the system has significant inertia: it takes a long time



from a cut in man-made greenhouse gas emissions to an observable lowering of the temperature. In fact, there is a fifty year delay in the base run from the peak emissions in 2030 to the peak in temperature in 2085. Second, the declining temperature trend in the 2090s does not signal the end to global warming in ESCIMO. This cooling is the net result of four simultaneous processes: the increasing use of atmospheric heat to melt snow and ice, the rapid decline in man-made $CO_2$-emissions, the

rising release of $CH_4$ from permafrost, and the darkening of the surface as there is more open ocean and exposed rock when snow and ice melts. For a short while, around the turn of the next century, the first two (cooling) processes overpower the latter two (warming) processes. But in the longer run, this is not always the case, as we discuss in the section on run-away behavior later in the paper.

### 3.5 Experimenting with ESCIMO

In the following pages, we will describe a number of experiments with ESCIMO where we simulate the effect of various possible human interventions and natural events.

The intervention is normally introduced in model year 2015 and the simulation run to 2100. The model output is then compared to the base run in order to measure the effect of the intervention, and to explain why the effect arises in the model system.

### 3.5.1 Coherent experiments

When making experiments with causal models such as ESCIMO, it is essential to restrict oneself to "coherent experiments". This means to restrict oneself to parameter changes that do not violate a) physical law or b) the requirement that the parameter set that is used in the experiment also should be able to reproduce history from 1850 to 2015.

Still, we do sometimes make incoherent parameter changes just in order to learn more about the dynamic properties of the model system. We have already shown one example in Fig. 8, when we tested the effect of an unrealistic sudden drop to zero

of man-made greenhouse gas emissions.

### 3.5.2 Uncertainty

Like all models, ESCIMO is a simplification of the "real world" and hence a model generated scenario for the future will deviate from the "real world future" – which will only be known after the fact. It is difficult to say anything meaningful and short about the precision level in the ESCIMO output. We know that the model is capable of reproducing history with the

precision level that can be inferred from Figures 5, 6, and 7, and that ESCIMO is able to reproduce the output of more complex models with the precision level which can be seen in Figures 7 and 8. It does not appear useful to calculate the $R^2$ of these fits, but it may be useful to say the fit is "of the order of ±20%". By this, we mean that the difference between the model generated time series and the reference time series is larger than ±10% and much smaller than ±100% on average. Similarly, it is difficult to say anything specific about the uncertainty in the change in curve shapes and in other marginal effects of an

intervention. Finally, it should be remembered that the historical data that are used to evaluate ESCIMO and all other models



are themselves uncertain. Even the most important variable – GMST – is not sharply defined. Several different competing time series purport to indicate what actually happened to temperature over the last 150 years (Hansen et al., 2010).

### 3.5.3 Very sensitive parameters

Some of the parameters in ESCIMO are sensitive in the sense that small changes in the parameter leads to big changes in
model output. The "true" value of such parameters is often unknown, all one knows from measurements is that the value lies within a certain uncertainty range. A "very sensitive" parameter is one that creates big changes in model output when it is varied within its uncertainty range. We use such very sensitive parameters when we fine-tune the model, i.e. when we try to optimize the model fit to historical data. Examples of two very sensitive parameters in ESCIMO are a) the parameter that describes the amount of outgoing long wave radiation that was blocked by water molecules in the atmosphere in 1850, and b)
the slope of the rising relation between the blocking and the concentration of water vapor. It is important to increase the knowledge about the numerical value of these sensitive parameters.

### 3.6 Simulating possible human interventions

As mentioned above, we restrict ourselves to coherent experiments – parameter sets that follow physical law and are able to recreate history. In addition we constrain ourselves to interventions that are not completely unrealistic politically speaking. In
this section, we describe the effect in ESCIMO of possible human responses to global warming, including geoengineering. In the following section, we describe the effect in ESCIMO of conceivable natural events, including disasters.

The result is summarized in Fig. 9, which shows the effect on GMST of six policy experiments and seven natural events – all introduced from 2015. The temperature path in the base run (dark blue) is included as a reference. In order for the effects of the various interventions to be comparable, we have tried to make them cost approximately the same, namely 1 T\$/yr. 1T
equals 1 trillion, which is $10^{12}$.





**Figure 9** **The effect in ESCIMO on the global average surface temperature (top) and sea level rise (bottom) of various possible policy interventions (left) and various natural events (right). All changes introduced in year 2015. World 1850 to 2100.**



Our choice of interventions is inspired by current discussions about global climate policy, for example ongoing attempts to reduce man-made emissions of greenhouse gases through less use of fossil fuels and less climate-intensive agriculture and forestry. The ambition level for such cuts has gradually increased during the series of Conference of Parties – COPs – following the United Nations Framework Convention on Climate Change in 1992. Presently, the ambition is to cut emissions by 40 %

in 2030 and 50-80 % in 2050, and to keep global warming below +2°C. The COP 21 in Paris in 2015 even agreed to seek to limit global warming to 1.5°C, although it was well known that this will require cuts way beyond what was achieved during the preceding 23 years.

We also discuss more exotic solutions to global warming. The Royal_Society (2009 ) provided a rather complete overview of the field, and deeper analyses of various options followed. Examples are Tokimatsu et al. (2015) who investigated zero

emissions scenarios. Geoengineering in the sense of making land surface brighter by brightening urban roofs was discussed by Gaffin et al. (2012) and placing mirrors over large stretches of desert areas by Jamieson (2013). A seminal paper on carbon capture and storage (CCS) was written by Haszeldine (2009), describing its potential. Holtsmark (2012) discussed the logging of boreal forest to produce biofuel, pointing out that this will not lead to reduced $CO_2$ emissions for at least a century – once the $CO_2$ is absorbed in new trees, which grow very slowly in the boreal ecosystem.

In the following sections, we discuss the interventions and events one by one.

### 3.6.1 Reducing GHG emissions by one third by 2035

In the base run, man-made GHG emissions peak in 2030 at 67 GtCO₂e/yr – or 18 GtCe/yr[2].  Emissions in the base run in 2050 are 14 GtCe/yr, about the same as in 2015, and decline to near zero in 2100. As mentioned above, strong voices are calling for faster cuts. An example would be to reduce emissions in 2035 by 5 GtCe/yr or around one third.

Such cuts can be achieved through investments in more energy efficiency, more renewable energy capacity, more carbon capture and geological storage (CCS), and reduced emissions from agriculture, forestry and waste. The cost of such cuts have been estimated to be 1 – 2 % of world GDP (UNEP, 2011). This very roughly amounts to 1 T$/yr, since the world GDP was 77 T$/yr in 2015.[3]

In Fig. 9 (left, black curve), we show the effect in ESCIMO of reducing man-made GHG emissions by 5 GtCe/yr in  2035,

increasing  linearly from 0  in 2015, and remaining at that level. In this scenario man-made emissions will still peak in 2030 and go to zero in 2080. The peak temperature in ESCIMO however shifts from +2.4°C around 2085 to +1.9°C around 2070. Figure 9 also shows that this policy reduces the sea level rise in 2100 from 0.9 to 0.7 meters relative to preindustrial time. But the sea level keeps rising for hundreds of years after 2100, even after the halt in man-made greenhouse gas emissions in 2100

---

[2] 1 GtCO₂e/yr equals one billion ($10^9$) tons of $CO_2$ equivalents per year. 1 GtC/yr equals one billion ($10^9$) tons of carbon per year. 1 tCe contains as much carbon as 3,7 tCO₂e.

[3] 1 T$ equals 1 trillion ($10^{12}$) $. The average cost per ton of carbon reduced would be equal to (1T$/yr)/(5GtCe/yr) = 200$/tCe or some 54$/tCO₂e.





In conclusion, cutting emissions by one third achieves the globally agreed goal of keeping warming below +2.0°C. An interesting question now becomes: Can other – and ideally cheaper – human interventions achieve the same result? What other interventions can be bought for 1 T$/yr? The next sections provide some answers.

### 3.6.2 Large scale implementation of carbon capture and geological storage (CCS)

An alternative use of 1 T$/yr would be to build and operate carbon capture and storage (CCS) plants on big point uses of coal, oil and gas across the world. An average CCS plant is able to remove 1 $MtCO_2$/yr from the smokestack emissions, compress it, and store it deep in Earth's crust. It costs around 2 G$ to build and run a CCS plant for its lifetime of 20 years. Thus, our available budget of 1 T$/yr will allow the continuing removal of 10 $GtCO_2$/yr.[4] If we start spending 1 T$ per year in 2015, we will reach full implementation in 2035, having built by then 10,000 CCS plants. Further investment will only compensate for
the closure of outdated CCS plants. The 10,000 CCS plants will remove 10 $GtCO_2$/yr, or 3 GtC/yr, thereafter, as long as the investment program is maintained, and as long as there is $CO_2$ to be removed.

Figure 9 (left, black dotted curve) shows that large scale implementation of CCS shifts the peak temperature in ESCIMO from +2.4°C in 2085 to +2.2°C in 2070. This means that using our budget of 1 T$/yr on CCS is less effective than using it on the general mitigation explored in point a above. This is self-evident given that reducing emissions using CCS is nearly twice as
expensive as the average of all the mitigation strategies used in section 3.6.1. The advantage with CCS is that it is a technological solution that can be retrofitted on big point sources after they are built, and furthermore that they can be used to remove $CO_2$ from the atmosphere once global society decides to do so, by burning biomass and collecting the $CO_2$.

### 3.6.3 Stopping tropical deforestation

The tropical forests, ca. 16 $Mkm^2$ are largely found in Amazonas, Congo and Indonesia (FAO, 2014). These forests contain a
significant stock of carbon, some 260 GtC (FAO, 2014), in the form of biomass in standing trees, dead trees, litter and soil – on average $260GtC/16Mkm^2$=16.000 $tC/km^2$. Of this, two thirds are above the ground and one third below. Perhaps one half of this carbon stock is removed when an area is "deforested" – that is logged and burned in preparation for alternative uses, like grazing, plantations, or urban development. Since the year 2000, the world has lost ca 1 % of its tropical forest per year (Martin, 2015) through land clearing. This means that the tropical forest at the start of this century lost an area of 16
$Mkm^2$*1%/yr = 160,000 $km^2$/yr, and carbon at the rate of perhaps 160,000 $km^2$/yr*8,000 tC/yr = 1.3 GtC/yr.

Figure 9 (left, light green curve) shows the effect of bringing tropical deforestation to a complete halt in 2015. This reduces the flow of $CO_2$ into the atmosphere, which cools, but at the same time darkens the surface (because forests are darker than grasslands), which warms. The net effect is less cooling than one might have expected. The cost of doing so would be miniscule compared to our budget of 1 T$/yr, even if done in the most expensive manner possible: by buying the land and protecting it
and its indigenous population. Assuming that the land can be bought for 100,000 $/km^2$, which is the cost of a productive

---

[4] $((1T\$/yr)/(2G\$/20yr))*1MtCO_2/yr = 10\ GtCO_2/yr$



Scandinavian forests, the annual cost would be 160.000 km$^2$/yr*100.000$/km$^2$ = 16 G$/yr, which is less than 2 % of the available budget.

But Figure 9 shows that although cheap, stopping tropical deforestation does not solve the problem. Cutting deforestation to zero only reduces the peak temperature by 0.3°C, because of the combined cooling and warming effect. In other words,

protection of existing tropical forests can not by itself solve stop global warming. Still, stopping tropical deforestation is crucial, because of its role in protecting global biodiversity, which resides primarily in the world's forests and coral reefs.

### 3.6.4 Stopping logging in Northern forests

The temperate and boreal forests, ca. 11 Mkm$^2$ (FAO, 2014) are largely found in North America, Europe and Russia. These forests contain a significant stock of carbon, some 215 GtC (FAO, 2014), in the form of biomass in standing trees, dead trees,

litter and soil – on average 215GtC/11Mkm$^2$=20,000 tC/km$^2$. Of this one quarter above the ground, much less than the two thirds above the ground in tropical forests. The "Northern" forests differ from the tropical forest: they do regrow after logging, although it takes a long time – a hundred years or so. The total forest area has been essentially stable over the last several decades, in spite of an average logging of 1,200 Mm$^3$/yr out of a total growing stock of some 120,000 Mm$^3$ in the first decade of this century (FAO, 2014). This amounts to[5] a removal of 0.12 GtC/yr = 1,200Mm$^3$/yr*0.1tC/m$^3$. But at the same time, the

Northern forests grew 2,400 Mm$^3$/yr, equivalent to binding 0.24GtC/yr = 2,400Mm$^3$/yr*0.1tC/m$^3$, filling in the open spaces left by the major clear-felling programs in the last half of the 1900s. The Northern forests currently act as a net sink, absorbing some 0.12 GtC/yr.

If the logging of the Northern forest were stopped, it would increase the rate of accumulation of $CO_2$ in the Northern forest to 0.24 GtC/yr. This would last until the forest was mature, in a century or so. At that time net accumulation would be close to

zero, because in a mature forest the annual formation of new wood through photosynthesis is more or less balanced by the rotting of an equal amount of wood. In the very long run – hundreds of years - there is a very slow accumulation of carbon in the soil.

Figure 9 (left, dark green curve) shows the effect of stopping logging of the Northern forest from 2015. This reduces the flow of $CO_2$ into the atmosphere, which cools, but at the same time darkens the surface (because trees are darker than clear cuts),

which warms. The net effect is less cooling than one might have expected.

The cost of stopping all logging would be low compared to our available budget of 1T$/yr. The price paid to forest owners for roundwood is some 40 $/m$^3$. Thus the value of the annual logging is (1,200Mm$^3$/yr)*40$/m$^3$ = 48 G$/yr, or less than 5 % of the budget.

---

[5] One living m$^3$ of spruce and pine weighs ca 0,4 tons. One half of this is water (0,2 tons) and the rest is dry matter (0,2 tons). One half of the dry matter is carbon (0,1 tons). Hence one m$^3$ of wood contains 0,1 tons of carbon irrespective of its humidity.



In summary, the net cooling effect of stopping logging of the Northern forest is small. Figure 9 shows that it amounts to less than one-half of the effect of stopping tropical deforestation. The value of a logging ban in pristine or old-growth Northern forests arises more from its positive impact on biodiversity and esthetics.

It is important to note that stopping the use of wood would require substitutes for both paper, construction materials and
bioenergy. The substitutes would lead to some emissions and reduce – but not eliminate – the net cooling effect of a ban on logging.

### 3.6.5 Making land surfaces brighter

A fourth alternative would be to use 1 T\$/yr to brighten urban areas with white paint, to increase the reflection from deserts using reflective sheeting, or to plant fields and grasslands with brighter crops (Irvine and Ridgwell, 2009). Using such
techniques it would be possible to increase the land surface albedo by 0.1 on average for the areas involved (Akbari et al., 2012) – and by 0.4 through reflective sheeting in hot deserts (Royal_Society, 2009 , p. 25). There exist much suitable land: the total land surface is 148 Mkm$^2$, of which urban land is ca 0.5 Mkm$^2$ and hot deserts ca 25 Mkm$^2$. The cost of painting and maintaining surfaces white is around 0.3 \$/m$^2$yr (Royal_Society, 2009 , p. 25). Thus our budget of 1 T\$/yr would suffice to brighten 3 Mkm$^2$ – that is some 2 % of the land surface (Royal_Society, 2009 ).
Figure 9 (left, yellow curve) shows that increasing the albedo by 0.1 on 2 % of the total land surface has a strong effect in ESCIMO. It shifts the peak in temperature from +2.4°C around 2085 to +1.7°C around 2070 when the whitening is fully implemented. The problem with this geoengineering strategy, however, is that no one knows what will be the local effects on temperature, precipitation, and wind from painting large areas white.

### 3.6.6 Injecting stratospheric aerosols

Many other ways have been proposed to increase the reflection of incoming solar energy. The most realistic of these "solar radiation management" techniques appears to be to increase the reflectivity of marine clouds (Irvine & Ridgwell, 2009, p. 154) by spraying their tops with aerosols (e.g. SO$_2$ from airplanes flying at 20.000 m altitude). Adding 1 – 5 MtS/yr to the stratosphere in this way is expected to be able to reduce incoming solar radiation by up to 4 W/m2. The cost is estimated to be between 3 and 30 \$/tS (Royal Society, 2009, p 29-32). Thus the cost of countering the warming from preindustrial times
through injection of aerosols in the stratosphere would be less than 150 G\$/yr, that is a fraction of our budget of 1 T\$/yr. However, the known unintended side effects of such large scale injection are huge, and in order to uncover the unknown side effects in a controlled manner, very gradual implementation would be needed.

Figure 9 (left, brown curve) shows the effect in ESCIMO of scattering back 3 W/m$^2$ to space from the top of stratospheric clouds, starting in 2015. This amounts to reflecting some 3/340 = 0.9 % of the incoming energy from the sun. The GMTS is
lowered relative to the base run by 1.9 °C in 2050 and by 2.7 °C in 2100. Figure 9 shows that it is possible to keep global warming well below +1°C throughout this century by injecting a small amount of aerosols in the stratosphere. But there would be serious unintended effects: The oceans would become increasingly acidic, as would downpour, and if the spraying



terminates, for instance because of war, the global temperature would immediately make a big jump upwards. The ethical and legal implications are also intricate, so few recommend this intervention. The only acceptable defense for large-scale injection might be to avoid a temporary peak in the temperature, while humanity was working to lower man-made emissions in other ways.

### 3.6.7 Human interventions - comments

The main conclusion from these experiments with possible human interventions is first that it is indeed possible to reduce global warming in this century. Second, that it is relatively inexpensive. If global society is willing to spend 1 % of world GDP to reduce global warming, it is possible to lower the temperature by up to 0.5°C in 2050 and up to 1.0°C in 2100 compared to the base run – if conventional interventions are used singly or in combination.  Third, it appears possible to lower the sea level rise in 2100 by up to 20 cm – so that we get a rise in this century of 0.4 m instead of 0.6 m. But here the main advantage will be during the next century and later, in the form of less thermal expansion. Fourth, conventional human interventions appear capable of keeping global warming below +2°C in this century, without resorting to geoengineering methods. (It is another question what will happen after 2100, see a later section.)  But it will require some 1 % of world GDP, which means the same as shifting 1 % of all labor and capital from dirty to clean production. Geoengineering such as stratospheric aerosols and brightening of the surface may be cheaper, but are likely to have major unintended and undesirable "side" effects.

It will be interesting to see whether these conclusions from ESCIMO are supported by experiments using more complex climate models.

### 3.7 Simulating possible natural events

Finally, we explore the effect of a number of possible natural events or disasters. We do this partly to trigger – and to study – additional dynamic behavior modes of ESCIMO, and partly to get a better feel for the sensitivity of the model system. In some of the experiments, it is difficult to stick to our ambition of coherence. For example, if one wants to test what will happen if the Greenland ice sheet melts twice as fast as in the base run, one must remember that this will require a huge amount of additional heat, at least compared to the heat content of the atmosphere. If one wants to be realistic, one cannot simply assume that suddenly one day, half of the ice is gone. All one can do is to assume an increase in the rate of transfer of heat from the air to the ice. However, the transfer rate is limited by physical law, to numbers well below what it takes to melt Greenland's ice sheet in a short time. Thus, to stay coherent, we have to assume another mechanism, for example that the Greenland ice sheet actually *slides* into the ocean due to increased lubrication of the ice/land interface and melts in the ocean. This would have the effect of cooling the ocean water – and therefore slow down global warming. Similar causal thinking was necessary in other experiments.

The right hand side of Fig. 9 shows the effect on the GMST and the sea level rise of a number of natural events, which are all introduced from model year 2015. The temperature and sea level in the base run (blue curves) are included as baselines.



### 3.7.1 Doubling the rate of melting of ice and snow

Figure 9 (right, black curve) shows the effect in ESCIMO of doubling the rate of melting (measured in Giga-tons of ice per year) of Arctic sea-ice, on-land glaciers, permafrost, and Greenland ice. We do this by doubling the melting rate at any temperature, implying a doubling of the rate of energy transfer from air to snow, ice and frozen soil at any given temperature.

Interestingly, this increases the temperature (by +1.1°C) in 2100 relative to the base run. The reason is that the faster melting of Arctic sea-ice, on-land glaciers and permafrost exposes more of the dark underlying ocean and rock, which in turn absorbs more incoming sunlight. The resulting heat accelerates the melting of the permafrost, which adds methane to the atmosphere and producing even higher temperatures. The Greenland ice contributes little to the albedo change, because the ice is so thick that it takes centuries before its area is reduced significantly.

### 10   3.7.2 Twice as many volcanoes, forest fires, and sunspots

The right hand side of Fig. 9 also shows the effect in ESCIMO of doubling the historical frequency of volcanic eruptions, forest fires, and sunspot activity, all of which are physically possible although unlikely.

Double volcanic activity (right, black dotted curve) leads to slight cooling in ESCIMO, approximately -0.3°C, because of the cooling effect of the increased amount of aerosols emitted. Double forest fires (right, brown curve) lead to slight warming

(+0.1°C) as a net effect of warming from more $CO_2$ and cooling from more smoke (aerosols). Doubling the sunspot activity (yellow curve) has no discernable effect in ESCIMO. The yellow curve winds around the base run (blue curve).

### 3.7.3 One percent more clouds – low and high

Finally, Fig. 9 shows the effect in ESCIMO of increasing the extent of low (cooling) clouds by 1 % artificially in 2015, and – separately – the extent of high (warming) clouds. We use such a small change because we know the model system is very

sensitive to changes in the low cloud cover. And more importantly, we know that this sensitivity is unrealistic and arises because the model is lacking most of the feedbacks from increased cloud cover back to the rest of the model system. These feedbacks should be included in an improved version of ESCIMO, probably in the form of a water sector where $H_2O$ is conserved.

More low clouds (right, light green curve) lead to significant cooling in ESCIMO (-0.7°C) – because the low clouds reflect the

incoming sunlight that hit them from above, more than they reflect the upwards heat radiation from the Earth's surface that hit them from below. More high clouds (orange curve) lead to some warming (+0.3°C).

### 3.7.4 Greenland ice sliding into the ocean

There is a worry that the Greenland ice sheet might slide into the ocean as a result of better lubrication of the ice/rock interface by flows of meltwater from the ice surface (Mouginot et al., 2015). Currently Greenland is losing 400 Gt/yr of ice (IPCC,



2013, p. 319) out of her total mass of some 3 million Gt of ice. This amounts to $(400Gt/yr)/(3.000.000Gt) = 0.013\ \%/yr$, which if continued would remove one half of the Greenland ice in 7.500 years.

But the rate of ice loss has been accelerating, and as an experiment we chose to test the effect of assuming that one quarter of the ice mass slides into the ocean during the next one hundred years.

Sliding the Greenland ice into the ocean at such a rate leads to significant cooling in ESCIMO. The GMST declines by up to -0.4°C relative to the base run because of all the heat needed to melt the ice that slides into the ocean. Once the melting is over, the temperature quickly moves back to the level it has in the base run – and then in fact surpasses it, due to reduced white area for reflection. But accelerated melting also leads to a significant rise in the sea level, well above 2.5 meters by 2100. Needless to say, this is an unrealistic scenario.

**3.8 Experimentation - comments**

Our main conclusion from experiments with ESCIMO is that the model system is robust. It requires very significant human interventions or natural events to alter the temperature in 2050 by more than 0.5°C.  The only exception is the injection of aerosols in the stratosphere, but this policy has significant undesirable side effects. It will be interesting to see whether these general conclusions from ESCIMO are supported by experiments using more complex climate models.

**4 Conclusions**

**4.1 Insights from running ESCIMO**

Our experiments support these tentative conclusions:

1.  ESCIMO can be used to calculate in a few seconds the effects of a wide variety of policy alternatives and natural events. Furthermore, ESCIMO is so simple and transparent that it is possible to understand what is going on in the
model system. Both the effects and the explanation needs to be corroborated by similar experiments using complex climate models.

2.  The model system is relatively stable. It requires big parameter changes to make the model system deviate significantly from the base run. This applies primarily to aggregate measures, like the *global mean surface temperature*, the *area covered by snow and ice*, or the *amount of carbon stored in biomass*.

3.  Still it is possible to influence future temperatures through realistic human interventions. Moderately expensive interventions, such as annual use of 1% of world GDP or some 1 T$/yr, can lower the temperature in 2050 by up to 0.5°C, and the temperature in 2100 by up to 1.0°C, compared to the base run. Much lower temperatures can be achieved, especially in 2100, if global society is willing to spend more than 1 % of GDP on reducing emissions. The temperature can also be lowered through geoengineering, but with significant negative side effects.

4.  The effect of clouds on temperature appears to be very strong. The drivers of high (warming) clouds and low (cooling) clouds are not well known, and should be the focus of more research.



5. The heat capacity of the deep ocean is huge and dominates the heat flows in the long run. The transfer mechanisms are not well known.

6. The land surface albedo does not change enough in ESCIMO before 2100 to significantly affect global warming. In this time window, the main albedo effect is the darkening of the surface that result from the decrease in ice and snow

cover. In the longer run, desertification may increase reflectivity.

## 4.2 Run-away behavior

In a system dynamics perspective, it is interesting to note that ESCIMO contains numerous self-reinforcing feedback loops, but that none of them cause "run-away" exponential growth before 2100. The reason is that the gain around the self-reinforcing loops in Figure 1 stays below one during this century. Thus the loops, although positive, create s-shaped growth, not

exponential growth.

But in the longer run, things appear to be different. When ESCIMO is run further into the future, under some conditions, we see continuing increase in temperature long after man-made GHG emissions have been brought to zero. Apparently, this occurs because the GMST stays high enough to continue the melting of the permafrost, which leads to continuing additions of methane to the atmosphere, increased blocking of outgoing radiation, and higher surface temperatures. This process will of course cease

in the very long run once all the permafrost is gone, but in ESCIMO this takes thousands of years. The era of warming is prolonged by the fact that the Earth's surface systematically gets darker as all sea ice, snow, glaciers,  and permafrost melt. So, in summary, in ESCIMO we do see continuing warming, for hundreds if not thousands of years, under some conditions. Luckily, this "self-reinforcing" temperature rise occurs very slowly (a few tenths of a degree per century) and not exponentially, but more linearly. This phenomenon may have two explanations. One is that it actually reflects reality. The

other one, and much more likely, is that it reflects a weakness in ESCIMO connected to the lack of a complete water cycle. We will look further into this in a follow-up paper.

Our initial investigation suggest that water vapor plays a crucial role in this dynamic. Water vapor prevents the cooling that would otherwise have brought the temperature back to the preindustrial normal, when the concentration of $CO_2$ in the atmosphere starts to wane. Water vapor does so by maintaining a temperature that is high enough to maintain the rate of

melting of permafrost and thereby adding fresh methane to the atmosphere as long as there is any permafrost left. Water vapor manages this even when there is no  man-made $CO_2$ in the atmosphere, because water is such a strong greenhouse gas. In other words, the water vapor shifts the system from a $CO_2$-dominated to a water vapor-dominated system. If the concentration of water vapor is lifted to a higher level – irrespective of the reason – the vapor tends to remain in the atmosphere, because the saturation level is determined by the temperature and nothing else.

Regardless, the tentative results from ESCIMO should be explored further in more complex models.



### 4.3 Very slow return to normal

Once humanity has emitted greenhouse gases to the atmosphere and generated global warming, only two natural mechanisms work to undo the damage. These are absorption of carbon in plants and soil biomass, and transmission of carbon into the deep ocean and sediments. Both processes work constantly to reduce the $CO_2$ content of the atmosphere, back towards the

preindustrial level, but they both are very slow. It will take thousands of years to lower the concentration of $CO_2$ to preindustrial levels. Hence, it is a good approximation to say that once a molecule of $CO_2$ has been added to the atmosphere, it will stay there for more than a thousand years – although it spends some of that time as non-perennial part of a tree or plant.

### 5 Further research

The next step should be to compare the results from our experiments with ESCIMO with the results from the same experiments

done with more complex climate models, to discover important discrepancies and their causes. Furthermore, it would be useful to increase the precision level in the estimate of the most sensitive  parameters in ESCIMO. Next, ESCIMO should be expanded to include a full water cycle, so that the model treats water in a complete fashion alongside carbon, energy and albedo. Finally, we will continue our exploration of the phenomenon of long term warming that appears in ESCIMO.

### 6 Data and model availability

The ESCIMO model is freely available on the web from , which is the website for the book 2052 – A Global Forecast for the Next Forty Years (Randers, 2012), and users can make their own simulations in seconds.

The downloadable file contains the model equations and parameters line-by-line, with explanations and literature references for each line. All exogenous data is gathered in one EXCEL file, which must also be downloaded and placed in the same directory as the model file. The file allows the user to make his/her own simulation runs, as long as s/he has Vensim simulation

software, which is available for free from their web page (2015d) - you need the Vensim Model Reader. The model contains 50 mostly non-linear, differential equations (i.e. there are 50 non-trivial state variables) and 600 active equations. For display purposes, we have added another 400 variables. Simulation time is about 1 second. ESCIMO is one integrated model, run by hitting one key, and with no need to coordinate the running of separate submodules, which is a challenge in common Earth System Models.

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
