# Peer review of "A User-friendly Earth System Model of Low Complexity: The ESCIMO system dynamics model of global warming towards 2100"

_Earth System Dynamics, 2016_

## Short Comment (SC1) · 9 May 2016

The model equations and documentation can be downloaded as zip files from http://www.2052.info/escimo/

---

## Referee Comment (RC1) · Anonymous Referee #1 · 26 Jun 2016

The authors present ESCIMO, a dynamic earth system model focused on climate change. ESCIMO is designed to run on laptop computers in seconds so it can be used interactively in education and policy contexts. ESCIMO replicates recent climate history and the output of complex Earth System Models (ESMs). It is used to explore the response of the climate to various policies designed to limit future warming and climate change.

Overall, the ESCIMO model helps to fill a great need: providing scientists, policymakers, business leaders, educators, the media and the public at large with the ability to explore for themselves the dynamics of the climate and its response to policies to mitigate anthropogenic climate change. The model is ambitious and the analysis pre-

sented in the paper shows that it is capable of replicating the historical data for a range of important climate variables and the behavior of larger, more complex models when driven with similar assumptions for future emissions. At the same time, there is clear room for improvement in the model, the analysis and testing to build confidence in it, and the model presentation and interface so that it can be more easily understood and used.

ESCIMO builds on the widely-used C-ROADS climate model developed by MIT and ClimateInteractive.org (Sterman et al. 2012, cited in the references), and it is useful to compare the two since they are both designed to run in seconds or faster on ordinary laptops and to be used interactively by a wide range of audiences. Like C-ROADS and other EMICs or EMLCs (Earth system models of intermediate or low complexity), ESCIMO aggregates key state variables such as the CO2 and heat content of the atmosphere and oceans into a small number of compartments rather than providing the spatial resolution as found in AOGCMs or complex Earth system models. The higher level of aggregation allows the model to replicate the historical behavior of the climate and simulations of complex ESMs while retaining the ability to run in seconds.

Like C-ROADS, ESCIMO is a physics-based model containing a representation of the carbon cycle, the energy balance of the Earth including radiative forcing and heat fluxes between the atmosphere and ocean, and other key elements of the climate. ESCIMO adds additional structure not explicit in C-ROADS relating, for example, to surface albedo and clouds, while offering less detail in other areas. For example, ES-CIMO includes explicit compartments for carbon in fossil fuel reserves, the atmosphere, biomass, permafrost, the surface ocean, and the deep ocean. It does not distinguish between C in biomass and soils, as C-ROADS does, and it treats the deep ocean as a single compartment, while C-ROADS and many other EMICS represent the ocean as consisting of a number of layers to capture slow and imperfect mixing. C-ROADS aggregates carbon in soils and permafrost into a single compartment, while ESCIMO represents C in permafrost separately while aggregating soil carbon together with C in

living biomass. ESCIMO represents CO2, CH4, N2O and aggregated compartments for the "Kyoto" gases and "Montreal" gases, while C-ROADS represents more individual species of non-CO2 GHGs, each with its own budget, atmospheric stock, lifetime and impact on radiative forcing. ESCIMO also includes a representation of arctic sea ice cover and some additional geophysical detail. These features allow ESCIMO to capture more of the positive feedbacks that may contribute to sink saturation and enhanced CO2/CH4 emissions that can amplify the impact of anthropogenic emissions, and positive feedbacks such as the ice-albedo feedback that leads to arctic amplification (even though the model does not capture spatial detail).

The behavior of the ESCIMO model fits historical data well, including global mean surface temperature, CO2 and CH4 concentrations, sea level rise, arctic ice extent, heat flux to the ocean, radiative forcing and ocean pH.

The authors also compare the behavior of the model to the behavior of the CMIP5 model ensemble for RCP4.5. The fit is good. However, the authors should compare the behavior of ESCIMO to a much wider range of scenarios, including all the RCPs, to demonstrate that ESCIMO remains reasonable across a much broader set of assumptions for GHG emissions.

Curiously, in the "base" run showing model behavior through 2100 global GHG emissions peak around 2040 and fall nearly to zero by 2100. Such large declines in emissions would require significant policy interventions in nations around the world or dramatic technical breakthroughs in low carbon energy, or a dramatically lower rate of global economic (and perhaps population) growth. The Paris agreement reached at COP21, even if fully implemented, does not come close to such large emissions reductions (see e.g. https://www.climateinteractive.org/programs/scoreboard/). It is unusual to define a base run in such models that assumes such strong policy actions, actions that no nations have committed to make. That said, it should be relatively easy to run ESCIMO using alternative assumptions for future emissions that would still allow the ESCIMO to be compared to the behavior of other ESMs.

The policy and sensitivity experiments presented in the paper show the potential for the use of ESCIMO. These include various emissions pathways, as well as representations of carbon capture and sequestration, sulfate aerosol injection, and others. The model is also used to simulate the impact of alternative assumptions about insolation, cloud behavior, and so forth.

Technical issues:

Assessing model fit: I agree with the discussion on p. 22 that $R^2$ is not particularly useful as a measure of goodness of fit for the model, but the claim that it is sufficient to say that the fit is "or the order of $\pm 20\%$" is not appropriate. The authors should provide goodness of fit statistics in addition to the graphs in Figure 5 comparing simulated and actual behavior. Relevant goodness of fit statistics would include the Mean Absolute Error or Root Mean Square Error (MAE/RMSE) and measures of bias (systematic differences between the data and model; there are some, for example, the model is consistently low compared to the data for arctic sea ice extent).

Documentation: The authors have provided the model, which is written in the Vensim simulation language, enabling anyone to download the model, examine the equations and run the model using the free Vensim Model Reader. They also provide the output of the Argonne National Laboratory model documentation tool, which provides an hyperlinked annotated summary of the model and equation listing. This is excellent practice.

However, the model does not fully conform to the documentation standards for dynamic models: the HTML documentation of the model shows that 64% of the equations/parameters in the model do not include explanatory comments or other documentation, and that 8% of the variables do not appear in any view of the Vensim model. The Vensim software returns 371 errors when the dimensional consistency check is run. These must be corrected. The model diagrams in Vensim are laid out poorly, making it more difficult to understand the structure of the model. The model should be divided into more views, each named appropriately, and the diagram showing the

structure of each view should be laid out to be more readable. There should be a much better dashboard or cockpit with key parameters and policy levers available for sensitivity and policy testing, along with the graphs showing the key outputs.

---

## Short Comment (SC2) · 6 Sep 2016

We thank you for a very knowledgeable, insightful and useful review of the paper and model. We agree on many of the points made, and contest some of them. In the following, we have selected those of the referee's comments that need a follow up, and enclosed them in quotation marks before we make our own remarks. "ESCIMO builds on the widely-used C-ROADS climate model developed by MIT and ClimateInteractive.org." ESCIMO does not build on C-ROADS but was developed independently, and it is certainly useful to compare the two, as the reviewer does, since they are similar with regard to methodological platform and scope. "For example, ESCIMO includes explicit compartments for carbon in fossil fuel reserves, the atmosphere, biomass, permafrost, the surface ocean, and the deep ocean. It does not distinguish between C in biomass and soils, as C-ROADS does,.." ESCIMO does indeed distinguish between C in biomass and soils and has actually a rather detailed set of variables that represent various carbon reservoirs. "The authors also compare the behavior of the model to the behavior of the CMIP5 model ensemble for RCP4.5. The fit is good. However, the authors should compare the behavior of ESCIMO to a much wider range of scenarios, including all the RCPs, to demonstrate that ESCIMO remains reasonable across a much broader set of assumptions for GHG emissions." We have used the central RCP4.5 in Figure 7 and then compared our emission scenario with the four RCPs in Figure 8. We felt that a more elaborate investigation of the CMIP5 model ensemble would require too much space, but we are open on this point. "Curiously, in the "base" run showing model behavior through 2100 global GHG emissions peak around 2040 and fall nearly to zero by 2100....It is unusual to define a base run in such models that assumes such strong policy actions, actions that no nations have committed to make." Our base run emission scenario is based on the forecasts in the book by Randers, J.: 2052: A Global Forecast for the Next Forty Years, Club of Rome, Chelsea Green 5 Publishing 2012, as explained in ch. 3.4. It is not very different from RCP26 and RCP45 in figure 7, and it does represent our current beliefs. "Assessing model fit: I agree with the discussion on p. 22 that R2 is not particularly useful as a measure of goodness of fit for the model, but the claim that it is sufficient to say that the fit is "or the order of 20%" is not appropriate. The authors should provide goodness of fit statistics in addition to the graphs in Figure 5 comparing simulated and actual behavior. Relevant goodness of fit statistics would include the Mean Absolute Error or Root Mean Square Error (MAE/RMSE) and measures of bias (systematic differences between the data and model; there are some, for example, the model is consistently low compared to the data for arctic sea ice extent)." Agreed. We will include MAE and RMSE for goodness of fit, and mean error for bias for the base run and history. The reviewer now turns to the documentation and the model. "..the model does not fully conform to the documentation standards for dynamic models: the HTML documentation of the model

shows that 64% of the equations/parameters in the model do not include explanatory comments or other documentation, and that 8% of the variables do not appear in any view of the Vensim model. The Vensim software returns 371 errors when the dimensional consistency check is run. These must be corrected. The model diagrams in Vensim are laid out poorly, making it more difficult to understand the structure of the model. The model should be divided into more views, each named appropriately, and the diagram showing the structure of each view should be laid out to be more readable. There should be a much better dashboard or cockpit with key parameters and policy levers available for sensitivity and policy testing, along with the graphs showing the key outputs." The Vensim software returns 371 errors when the dimensional consistency check is run because some of our units are too complicated for the documentation tool to parse correctly. It does not mean that the units are inconsistent. We have corrected all unit 'errors' flagged by the documentation tool and will upload the corrected version shortly. We will also strip the diagram of all variables that are only there for experiments, and use different colors for the sectors. This will provide for a much better overview and readability.

---

## Referee Comment (RC2) · Anonymous Referee #2 · 12 Sep 2016

Review of the manuscript entitled "A User-friendly Earth System Model of Low Complexity: The ESCIMO system dynamics model of global warming towards 2100" by J. Randers et al.

The manuscript discusses the formulation of simple system dynamics model ESCIMO, Earth System Model of Low Complexity which can be run on laptop computers. This is a simpler and computationally inexpensive model and can be used by the policy makers and experts. The response of the climate to various policy interventions for reducing future global warming is discussed in the manuscript. The results presented in the paper are encouraging, but need further testing and analysis before it can be made available for making policy decisions. A through comparison of the model results

is needed, especially with all the RCP scenarios of CMIP5 is necessary to evaluate the model performance. The details about the base run, the forcings used for the base run and its performance is very much necessary. The forcings are discussed as "The simple ESCIMO model structure, when parameterized with plausible parameter values obtained from the literature or common sense, and driven by actual man made emissions of greenhouse gases from 1850 to 2015. . .The future portion of these graphs is generated by ESCIMO with what we see as the most likely man made emissions from 2015 to 2100", is not justifiable. The global mean fields shown for the base run (Fig. 5) show a steady increase till 2070 and then a decrease in temperature and other fields. The base run also show GHG emissions sharply decrease after 2040, all these needs clarification. The model performance matrix depends on the base run characteristics, which need refinement before assessing the policy interventions for reducing global mean temperature.

---

## Author Comment (AC1) · 26 Sep 2016

**Reply to the Interactive comments by the anonymous referees on**

**J. Randers et al.**

**"A User-friendly Earth System Model of Low Complexity: The ESCIMO system dynamics model of global warming towards 2100"**

We agree on many of the points made, and contest some of them. In the following, we have selected those of the referees' comments that need a follow up. We have placed the referees' comment in quotation marks before we make our own remarks.

**Anonymous Referee #1, published 26 June 2016.**
This is a very knowledgeable, insightful and useful review of the paper and model. We have numbered the comments for future reference.

1. "ESCIMO builds on the widely-used C-ROADS climate model developed by MIT and ClimateInteractive.org." ESCIMO does not build on C-ROADS: it was developed independently. But it is certainly useful to compare the two, as the reviewer does, since they are similar with regard to methodological platform and scope. *No changes are necessary to paper*.

2. "For example, ESCIMO includes explicit compartments for carbon in fossil fuel reserves, the atmosphere, biomass, permafrost, the surface ocean, and the deep ocean. It does not distinguish between C in biomass and soils, as C-ROADS does,.." ESCIMO does indeed distinguish between C in biomass and soils and has actually a rather detailed set of variables that represent various carbon reservoirs. *No changes are necessary to the paper*.

3. "The authors also compare the behavior of the model to the behavior of the CMIP5 model ensemble for RCP4.5. The fit is good. However, the authors should compare the behavior of ESCIMO to a much wider range of scenarios, including all the RCPs, to demonstrate that ESCIMO remains reasonable across a much broader set of assumptions for GHG emissions." We have used the central RCP4.5 in Figure 7 and then compared our emission scenario with four other RCPs in Figure 8. *No changes are necessary to the paper.*

4. "Curiously, in the "base" run showing model behavior through 2100 global GHG emissions peak around 2040 and fall nearly to zero by 2100….It is unusual to define a base run in such models that assumes such strong policy actions, actions that no nations have committed to make." Our base run emission scenario is based on the forecasts in the book by Randers, J.: 2052: A Global Forecast for the Next Forty Years, Chelsea Green Publishing, Vermont, 2012, as explained in ch. 3.4. The resulting base run is not very different from RCP2.6 and RCP4.5 in figure 7, and it does represent our current beliefs. *We will however include the main reasoning that explains the global GHG emissions scenarios that drives the base run through 2100. It is mainly determined by our forecast of population, GDP per person, and technological advance, and refer to the website www.2052.info for more detail.*

5. "Assessing model fit: I agree with the discussion on p. 22 that $R^2$ is not particularly useful as a measure of goodness of fit for the model, but the claim that it is sufficient to say that the fit is "or the order of 20%" is not appropriate. The authors should provide goodness of fit statistics in addition to the graphs in Figure 5 comparing simulated and actual behavior. Relevant goodness of fit statistics would include the Mean Absolute Error or Root Mean

Square Error (MAE/RMSE) and measures of bias (systematic differences between the data and model; there are some, for example, the model is consistently low compared to the data for arctic sea ice extent)." *Agreed. We will include MAE and RMSE for goodness of fit, and mean error for bias for the base run and history*.

The reviewer now turns to the documentation and the model.

6. "..the model does not fully conform to the documentation standards for dynamic models: the HTML documentation of the model shows that 64% of the equations/parameters in the model do not include explanatory comments or other documentation, and that 8% of the variables do not appear in any view of the Vensim model. The Vensim software returns 371 errors when the dimensional consistency check is run. These must be corrected. The model diagrams in Vensim are laid out poorly, making it more difficult to understand the structure of the model. The model should be divided into more views, each named appropriately, and the diagram showing the structure of each view should be laid out to be more readable. There should be a much better dashboard or cockpit with key parameters and policy levers available for sensitivity and policy testing, along with the graphs showing the key outputs." The Vensim software returns 371 errors when the dimensional consistency check is run because some of our units are too complicated for the documentation tool to parse correctly. It does not mean that the units are inconsistent. *We have corrected all unit 'errors' flagged by the documentation tool and will upload the corrected version shortly. We will also strip the diagram of all variables that are only there for experiments, and use different colors for different sub-types of variables, for example input data, lookup functions, etc.. This will provide for a much better overview and readability.*

**Anonymous referee #2, published: 12 September 2016**

7. "The results presented in the paper are encouraging, but need further testing and analysis before it can be made available for making policy decisions. A thorough comparison of the model results is needed, especially with all the RCP scenarios of CMIP5 is necessary to evaluate the model performance." *This has already been done as per §3 above. See also reply to §10 below.*
8. "The details about the base run, the forcings used for the base run and its performance is very much necessary." Here is an apparent misunderstanding: Carbon forcing is not an input to the model, but an output. *No changes are necessary to the paper*.
9. "The forcings are discussed as "The simple ESCIMO model structure, when parameterized with plausible parameter values obtained from the literature or common sense, and driven by actual man made emissions of greenhouse gases from 1850 to 2015. . .The future portion of these graphs is generated by ESCIMO with what we see as the most likely man made emissions from 2015 to 2100", is not justifiable. The global mean fields shown for the base run (Fig. 5) show a steady increase till 2070 and then a decrease in temperature and other fields. The base run also show GHG emissions sharply decrease after 2040, all these needs clarification." *We will justify this according to §4 above*.
10. "The model performance matrix depends on the base run characteristics, which need refinement before assessing the policy interventions for reducing global mean temperature." *The central idea behind making a simple model like ESCIMO is that any user with a PC can explore the model himself by entering his own beliefs about future carbon emissions. Our base run is based on our own beliefs, which are justified at very great length in the book by Randers, J.: 2052: A Global Forecast for the Next Forty Years, Chelsea Green Publishing, Vermont 2012 and on the website www.2052.info. No changes are necessary to the paper.*

---

## Author Comment (AC2) · 26 Sep 2016

The comment was uploaded in the form of a supplement:
http://www.earth-syst-dynam-discuss.net/esd-2016-13/esd-2016-13-AC2-supplement.zip